# NEURAL VARIATIONAL INFERENCE FOR EMBEDDING KNOWLEDGE GRAPHS

## ABSTRACT

Recent advances in Neural Variational Inference allowed for a renaissance in latent variable models in a variety of domains involving high-dimensional data. In this paper, we introduce two generic Variational Inference frameworks for generative models of Knowledge Graphs; Latent Fact Model and Latent Information Model. While traditional variational methods derive an analytical approximation for the intractable distribution over the latent variables, here we construct an inference network conditioned on the symbolic representation of entities and relation types in the Knowledge Graph, to provide the variational distributions. The new framework can create models able to discover underlying probabilistic semantics for the symbolic representation by utilising distributions which permit training by back-propagation in the context of neural variational inference, resulting in a highly-scalable method. Under a Bernoulli sampling framework, we provide an alternative justification for commonly used techniques in large-scale stochastic variational inference, which drastically reduces training time at a cost of an additional approximation to the variational lower bound. The generative frameworks are flexible enough to allow training under any prior distribution that permits application of the re-parametrisation trick, as well as under any scoring function that permits maximum likelihood estimation of the parameters. Experimental results display the potential and efficiency of this framework by improving upon multiple benchmarks with Gaussian prior representations. Code publicly available on Github.

## 1 INTRODUCTION

In many fields, including physics and biology, being able to represent *uncertainty* is of crucial importance (Ghahramani, 2015). For instance, when link prediction in Knowledge Graphs is used for driving expensive pharmaceutical experiments (Bean et al., 2017), it would be beneficial to know what is the confidence of a model in its predictions. However, a significant shortcoming of current neural link prediction models (Dettmers et al., 2017; Trouillon et al., 2016a) – and for the vast majority of neural representation learning approaches – is their inability to express a notion of uncertainty.

Furthermore, Knowledge Graphs can be very large and web-scale (Dong et al., 2014) and often suffer from incompleteness and sparsity (Dong et al., 2014). In a generative probabilistic model, we could leverage the variance in model parameters and predictions for finding which facts to sample during training, in an Active Learning setting (Kapoor et al., 2007; Gal et al., 2017).(Gal & Ghahramani, 2016) use dropout for modelling uncertainty, however, this is only applied at test time.

However, current neural link prediction models typically only return point estimates of parameters and predictions (Nickel et al., 2016), and are trained *discriminatively* rather than *generatively*: they aim at predicting one variable of interest conditioned on all the others, rather than accurately representing the relationships between different variables (Ng & Jordan, 2001), however, (Gal & Ghahramani, 2016) could still be applied to get uncertainty estimates for these models. The main argument of this article is that there is a lack of methods for quantifying predictive uncertainty in a knowledge graph embedding representation, which can only be utilised using probabilistic modelling, as well as a lack of expressiveness under fixed-point representations. This constitutes a significant contribution to the existing literature because we introduce a framework for creating a family of highly scalable probabilistic models for knowledge graph representation, in a field where there has been a lack of this. We do this in the context of recent advances in variational inference, allowing the use of any prior

distribution that permits a re-parametrisation trick, as well as any scoring function which permits maximum likelihood estimation of the parameters.

## 2 BACKGROUND

In this work, we focus on models for *predicting missing links* in large, multi-relational networks such as FREEBASE. In the literature, this problem is referred to as *link prediction*. We specifically focus on *knowledge graphs*, i.e., graph-structured knowledge bases where factual information is stored in the form of relationships between entities. Link prediction in knowledge graphs is also known as *knowledge base population*. We refer to Nickel et al. (2016) for a recent survey on approaches to this problem.

A knowledge graph $\mathcal{G} \triangleq \{(r, a_1, a_2)\} \subseteq \mathcal{R} \times \mathcal{E} \times \mathcal{E}$ can be formalised as a set of triples (facts) consisting of a relation type $r \in \mathcal{R}$ and two entities $a_1, a_2 \in \mathcal{E}$, respectively referred to as the *subject* and the *object* of the triple. Each triple $(r, a_1, a_2)$ encodes a relationship of type $r$ between $a_1$ and $a_2$, represented by the fact $r(a_1, a_2)$.

*Link prediction* in knowledge graphs is often simplified to a *learning to rank* problem, where the objective is to find a score or ranking function $\phi_r^\Theta : \mathcal{E} \times \mathcal{E} \mapsto \mathbb{R}$ for a relation $r$ that can be used for ranking triples according to the likelihood that the corresponding facts hold true.

### 2.1 NEURAL LINK PREDICTION

Recently, a specific class of link predictors received a growing interest (Nickel et al., 2016). These predictors can be understood as multi-layer neural networks. Given a triple $\mathbf{x} = (s, r, o)$, the associated score $\phi_r^\Theta(s, o)$ is given by a neural network architecture encompassing an *encoding layer* and a *scoring layer*.

In the encoding layer, the subject and object entities $s$ and $o$ are mapped to low-dimensional vector representations (embeddings) $\mathbf{h}_s \triangleq \mathbf{h}(s) \in \mathbb{R}^k$ and $\mathbf{h}_o \triangleq \mathbf{h}(o) \in \mathbb{R}^k$, produced by an encoder $\mathbf{h}^\Gamma : \mathcal{E} \to \mathbb{R}^k$ with parameters $\Gamma$. Similarly, relations $r$ are mapped to $\mathbf{h}_r \triangleq \mathbf{h}(r) \in \mathbb{R}^k$. This layer can be pre-trained (Vylomova et al., 2016) or, more commonly, learnt from data by back-propagating the link prediction error to the encoding layer (Bordes et al., 2013a; Nickel et al., 2016; Trouillon et al., 2016a).

The scoring layer captures the interaction between the entity and relation representations $\mathbf{h}_s$, $\mathbf{h}_o$ and $\mathbf{h}_r$ are scored by a function $\phi^\Theta(\mathbf{h}_s, \mathbf{h}_o, \mathbf{h}_r)$, parametrised by $\Theta$. Other work encodes the entity-pair in one vector (Riedel et al., 2013).

Summarising, the high-level architecture is defined as:

$$
\begin{aligned}
\mathbf{h}_s, \mathbf{h}_o, \mathbf{h}_r &\triangleq \mathbf{h}^\Gamma(s), \mathbf{h}^\Gamma(o), \mathbf{h}^\Gamma(r) \\
\phi(s, o, r) &\triangleq \phi^\Theta(\mathbf{h}_s, \mathbf{h}_o, \mathbf{h}_r)
\end{aligned}
$$

Ideally, more likely triples should be associated with higher scores, while less likely triples should be associated with lower scores.

While the literature has produced a multitude of encoding and scoring strategies, for brevity we overview only a small subset of these. However, we point out that our method makes no further assumptions about the network architecture other than the existence of an argument encoding layer.

### 2.2 ENCODING LAYER

Given an entity $e \in \mathcal{E}$, the entity encoder $\mathbf{h}^\Gamma$ is usually implemented as a simple embedding layer $\mathbf{h}^\Gamma(e) \triangleq [\Gamma]_e$, where $\Gamma$ is an embedding matrix (Nickel et al., 2016). For pre-trained embeddings, the embedding matrix is fixed. Note that other encoding mechanisms are conceivable, such as; recurrent, graph convolution (Kipf & Welling, 2016a;b) or convolutional neural networks (Dettmers et al., 2017).

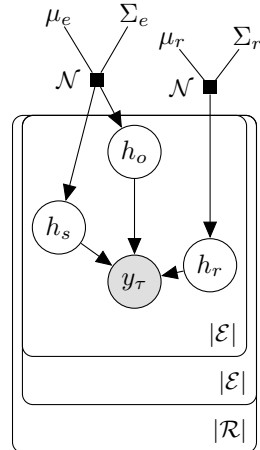

Figure 1: Latent Fact Model (LFM )

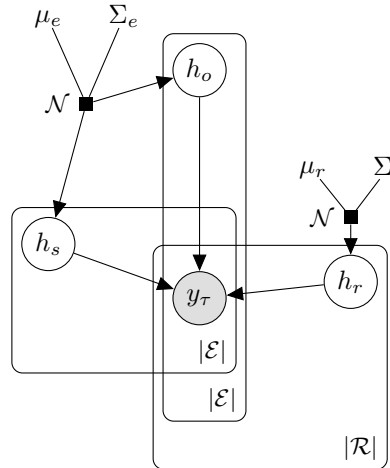

Figure 2: Latent Information Model (LIM)

### 2.3 DECODING LAYER: SCORING FUNCTIONS

**DistMult** DISTMULT (Yang et al., 2015) represents each relation $r$ using a parameter vector $\Theta \in \mathbb{R}^k$, and scores a link of type $r$ between $(\mathbf{h}_s, \mathbf{h}_o, \mathbf{h}_r)$ using the following scoring function:

$$\phi^{\Theta}(\mathbf{h}_s, \mathbf{h}_o, \mathbf{h}_r) \triangleq \langle \mathbf{h}_s, \mathbf{h}_o, \mathbf{h}_r \rangle \triangleq \sum_{i=1}^{k} \mathbf{h}_{s,i}\mathbf{h}_{o,i}\mathbf{h}_{r,i},$$

where $\langle \cdot, \cdot, \cdot \rangle$ denotes the tri-linear dot product.

**ComplEx** COMPLEX (Trouillon et al., 2016a) is an extension of DISTMULT using complex-valued embeddings while retaining the mathematical definition of the dot product. In this model, the scoring function is defined as follows:

$$\phi^{\Theta}(\mathbf{h}_r, \mathbf{h}_s, \mathbf{h}_o) \triangleq \text{Re}\left(\langle \mathbf{h}_r, \mathbf{h}_s, \overline{\mathbf{h}_o} \rangle\right),$$
$$\triangleq\, <\text{Re}\left(e_s\right), \text{Re}\left(e_r\right), \text{Re}\left(e_o\right)> + <\text{Im}\left(e_s\right), \text{Re}\left(e_r\right), \text{Im}\left(e_o\right)>$$
$$+\, <\text{Re}\left(e_s\right), \text{Im}\left(e_r\right), \text{Im}\left(e_o\right)> - <\text{Im}\left(e_s\right), \text{Im}\left(e_r\right), \text{Re}\left(e_o\right)>$$

where $\mathbf{h}_s, \mathbf{h}_o, \mathbf{h}_r \in \mathbb{C}^k$ are complex vectors, $\overline{\mathbf{x}}$ denotes the complex conjugate of $\mathbf{x}$, $\text{Re}\left(\mathbf{x}\right) \in \mathbb{R}^k$ denotes the real part of $\mathbf{x}$ and $\text{Im}\left(\mathbf{x}\right) \in \mathbb{C}^k$ denotes the imaginary part of $\mathbf{x}$.

## 3 GENERATIVE MODELS

Let $\mathcal{D} \triangleq \{(\tau_1, y_1), \ldots, (\tau_n, y_n)\}$ denote a set of labelled triples, where $\tau_i \triangleq \langle s_i, p_i, o_i \rangle$, and $y_i \in \{0, 1\}$ denotes the corresponding label, denoting that the fact encoded by the triple is either *true* or *false*. We can assume $\mathcal{D}$ is generated by a corresponding *generative model*. In the following, we propose two alternative generative models.

### 3.1 LATENT FACT MODEL

In Figure 1's graphical model, we assume that the Knowledge Graph was generated according to the following generative model. Let $\mathcal{V} \triangleq \mathcal{E} \times \mathcal{R} \times \mathcal{E}$ the space of possible triples. where $\tau \triangleq \langle s, p, o \rangle$, and $\mathbf{h}_\tau \triangleq [\mathbf{h}_s, \mathbf{h}_p, \mathbf{h}_o]$ denotes the sampled embedding representations of $s, o \in \mathcal{E}$ and $p \in \mathcal{R}$.

Note that, in this model, the embeddings are sampled for each triple. As a consequence, the set of latent variables in this model is $\mathcal{H} \triangleq \{\mathbf{h}_\tau \mid \tau \in \mathcal{E} \times \mathcal{R} \times \mathcal{E}\}$.

The joint probability of the variables $p^\theta(\mathcal{H}, \mathcal{D})$ is defined as follows:

$$p^\theta(\mathcal{H}, \mathcal{D}) \triangleq \prod_{(\tau, y_\tau) \in \mathcal{D}} p^\theta(\mathbf{h}_\tau)p^\theta(y_\tau \mid \mathbf{h}_\tau) \tag{1}$$

The marginal distribution over $\mathcal{D}$ is then bounded as follows, with respect to our variational distribution $q$:

$$p^\theta(\mathcal{D}) \geq \mathbb{E}_{q^\phi}\left[\log p^\theta(y_\tau \mid \mathbf{h}_\tau)\right] - \text{KL}[q^\phi(\mathbf{h}_\tau) \mid\mid p^\theta(\mathbf{h}_\tau)] \tag{2}$$

**Proposition 1** *As a consequence, the log-marginal likelihood of the data, under the Latent Fact Model, is bounded by:*

$$\log p^\theta(\mathcal{D}) \leq \sum_{(\tau, y_\tau) \in \mathcal{D}} \mathbb{E}_{q^\phi}\left[\log p^\theta(y_\tau \mid \mathbf{h}_\tau)\right] - \text{KL}[q^\phi(\mathbf{h}_\tau) \mid\mid p^\theta(\mathbf{h}_\tau)] \triangleq \text{ELBO} \tag{3}$$

*Proof.* We refer the reader to the Appendix 6 for a detailed proof LFM's ELBO.

**Assumptions:** LFM model assumes each *fact* of is a randomly generated variable, as well as a mean field variational distribution and that each training example is independently distributed.

### 3.1.1 OPTIMISING THE ELBO

Note that this is an enormous sum over $|\mathcal{D}|$ elements. However, this can be approximated via Importance Sampling, or Bernoulli Sampling (Botev et al., 2017).

$$\begin{aligned}
\text{ELBO} &= \sum_{(\tau, y_\tau) \in \mathcal{D}} \mathbb{E}_{q^\phi}\left[\log p^\theta(y_\tau \mid \mathbf{h}_\tau)\right] - \text{KL}[q^\phi(\mathbf{h}_\tau) \mid\mid p^\theta(\mathbf{h}_\tau)] \\
&= (\sum_{(\tau, y_\tau) \in \mathcal{D}^+} \mathbb{E}_{q^\phi}\left[\log p^\theta(y_\tau \mid \mathbf{h}_\tau)\right] - \text{KL}[q^\phi(\mathbf{h}_\tau) \mid\mid p^\theta(\mathbf{h}_\tau)]) \\
&+ (\sum_{(\tau, y_\tau) \in \mathcal{D}^-} \mathbb{E}_{q^\phi}\left[\log p^\theta(y_\tau \mid \mathbf{h}_\tau)\right] - \text{KL}[q^\phi(\mathbf{h}_\tau) \mid\mid p^\theta(\mathbf{h}_\tau)])
\end{aligned} \tag{4}$$

By using Bernoulli Sampling, ELBO can be approximated by:

$$\text{ELBO} \approx \sum_{(\tau, y_\tau) \in \mathcal{D}} \frac{s_\tau}{b_\tau}(\ \mathbb{E}_{q^\phi}\left[\log p^\theta(y_\tau \mid \mathbf{h}_\tau)\right] - \text{KL}[q^\phi(\mathbf{h}_\tau) \mid\mid p^\theta(\mathbf{h}_\tau)]\ ). \tag{5}$$

where $p^\theta(s_\tau = 1) = b_\tau$ can be defined as the probability that for the coefficient $s_\tau$ each positive or negative fact $\tau$ is equal to one (i.e is included in the ELBO summation). The exact ELBO can be recovered from setting $b_\tau = 1.0$ for all $\tau$. We can define a probability distribution of sampling from $\mathcal{D}^+$ and $\mathcal{D}^-$ – similarly to Bayesian Personalised Ranking (Rendle et al., 2009), we sample one negative triple for each positive one — we use a constant probability for each element depending on whether it is in the positive or negative set.

**Proposition 2** *The Latent Fact models* ELBO *can be estimated similarly using a constant probability for positive or negative samples, we end up with the following estimate:*

$$\begin{aligned}
\text{ELBO} \approx (&\sum_{(\tau, y_\tau) \in \mathcal{D}^+} \frac{s_\tau}{b^+}(\ \mathbb{E}_{q^\phi}\left[\log p^\theta(y_\tau \mid \mathbf{h}_\tau)\right] - \text{KL}[q^\phi(\mathbf{h}_\tau) \mid\mid p^\theta(\mathbf{h}_\tau)]\ )) \\
+ (&\sum_{(\tau, y_\tau) \in \mathcal{D}^-} \frac{s_\tau}{b^-}(\ \mathbb{E}_{q^\phi}\left[\log p^\theta(y_\tau \mid \mathbf{h}_\tau)\right] - \text{KL}[q^\phi(\mathbf{h}_\tau) \mid\mid p^\theta(\mathbf{h}_\tau)]\ ))
\end{aligned} \tag{6}$$

where $b^+ = |\mathcal{D}^+|/|\mathcal{D}^+|$ and $b^- = |\mathcal{D}^+|/|\mathcal{D}^-|$.

### 3.2 LATENT INFORMATION MODEL

In Figure 2's graphical model, we assume that the Knowledge Graph was generated according to the following generative model. Let $\mathcal{V} \triangleq \mathcal{E} \times \mathcal{R} \times \mathcal{E}$ the space of possible triples.

Similarly, $\tau \triangleq \langle s, p, o \rangle$, and $\mathbf{h}_\tau \triangleq [\mathbf{h}_s, \mathbf{h}_p, \mathbf{h}_o]$ denotes the sampled embedding representations of $s, o \in \mathcal{E}$ and $p \in \mathcal{R}$. The set of latent entity variables in this model is $\mathcal{H}_e \triangleq \{\mathbf{h}_e \mid e \in \mathcal{E}\}$ and the set

of latent predicate variables $\mathcal{H}_\mathrm{p} \triangleq \{\mathbf{h}_\mathrm{p} \mid \mathrm{p} \in \mathcal{R}\}$. With $\mathcal{H} = \mathcal{H}_e \cup \mathcal{H}_\mathrm{p}$. The joint probability of the variables $p^\theta(\mathcal{H}, \mathcal{D})$ is defined as follows:

$$p^\theta(\mathcal{H}, \mathcal{D}) \triangleq \prod_{e \in \mathcal{E}} p^\theta(\mathbf{h}_e) \prod_{\mathrm{p} \in \mathcal{R}} p^\theta(\mathbf{h}_\mathrm{p}) \prod_{(\tau, y_\tau) \in \mathcal{D}} p^\theta(y_\tau \mid \mathbf{h}_\tau) \tag{7}$$

The marginal distribution over $\mathcal{D}$ is then defined as follows:

$$p^\theta(\mathcal{D}) \triangleq \int \prod_{e \in \mathcal{E}} p^\theta(\mathbf{h}_e) \prod_{\mathrm{p} \in \mathcal{R}} p^\theta(\mathbf{h}_\mathrm{p}) \prod_{(\tau, y_\tau) \in \mathcal{D}} p^\theta(y_\tau \mid \mathbf{h}_\tau) d\mathcal{H} \tag{8}$$

**Proposition 3** *The log-marginal likelihood of the data, under the Latent Information Model, is the following:*

$$\log p^\theta(\mathcal{D}) \geq \mathbb{E}_{q^\phi} \left[ \log p^\theta(\mathcal{D} \mid \mathcal{H}_e, \mathcal{H}_\mathrm{p}) \right] - \mathrm{KL}[q^\phi(\mathcal{H}_e) \mid\mid p^\theta(\mathcal{H}_e)] - \mathrm{KL}[q^\phi(\mathcal{H}_\mathrm{p}) \mid\mid p^\theta(\mathcal{H}_\mathrm{p})] \tag{9}$$

*Proof.* We refer the reader to the Appendix 6 for a detailed proof LIM's ELBO.

**Assumptions:** LIM assumes each variable of *information* is randomly generated, as well as a mean field variational distribution and that each training example is independently distributed. This leads to a factorisation of the ELBO that seperates the KL term from the observed triples, making the approximation to the ELBO through Bernoulli sampling simpler, as the KL term is no longer approximated and instead fully computed.

### 3.2.1 Optimising the ELBO

Similarly to Section 3.1.1, by using Bernoulli Sampling the ELBO can be approximated by:

$$\mathrm{ELBO} \approx \left( \sum_{(\tau, y_\tau) \in \mathcal{D}} \frac{s_\tau}{b_\tau} \mathbb{E}_{q^\phi} \left[ \log p^\theta(y_\tau \mid \mathbf{h}_\tau) \right] \right) - \left( \sum_{e \in \mathcal{E}} \mathrm{KL}[q^\phi(\mathbf{h}_e) \mid\mid p^\theta(\mathbf{h}_e)] \right) \\ - \left( \sum_{\mathrm{p} \in \mathcal{R}} \mathrm{KL}[q^\phi(\mathbf{h}_\mathrm{p}) \mid\mid p^\theta(\mathbf{h}_\mathrm{p})] \right). \tag{10}$$

Which can be estimated similarly using a constant probability for positive or negative samples, we end up with the following estimate:

**Proposition 4** *The Latent Information Models* ELBO *can be estimated similarly using a constant probability for positive or negative samples, we end up with the following estimate:*

$$\mathrm{ELBO} \approx \left( \sum_{(\tau, y_\tau) \in \mathcal{D}^+} \frac{s_\tau}{b^+} \mathbb{E}_{q^\phi} \left[ \log p^\theta(y_\tau \mid \mathbf{h}_\tau) \right] \right) + \left( \sum_{(\tau, y_\tau) \in \mathcal{D}^-} \frac{s_\tau}{b^-} \mathbb{E}_{q^\phi} \left[ \log p^\theta(y_\tau \mid \mathbf{h}_\tau) \right] \right) \\ - \left( \sum_{e \in \mathcal{E}} \mathrm{KL}[q^\phi(\mathbf{h}_e) \mid\mid p^\theta(\mathbf{h}_e)] \right) - \left( \sum_{\mathrm{p} \in \mathcal{R}} \mathrm{KL}[q^\phi(\mathbf{h}_\mathrm{p}) \mid\mid p^\theta(\mathbf{h}_\mathrm{p})] \right) \tag{11}$$

where $b^+ = |\mathcal{D}^+|/|\mathcal{D}^+|$ and $b^- = |\mathcal{D}^+|/|\mathcal{D}^-|$.

## 4 Related Work

Variational Deep Learning has seen great success in areas such as parametric/non-parametric document modelling (Miao et al., 2017; Miao et al., 2016) and image generation (Kingma & Welling, 2013b). Stochastic variational inference has been used to learn probability distributions over model weights (Blundell et al., 2015), which the authors named "Bayes By Backprop". These models have proven powerful enough to train deep belief networks (Vilnis & McCallum, 2014), by improving upon the stochastic variational bayes estimator (Kingma & Welling, 2013b), using general variance reduction techniques.

Previous work has also researched word embeddings within a Bayesian framework (Zhang et al., 2014; Vilnis & McCallum, 2014), as well as researched graph embeddings in a Bayesian framework

(He et al., 2015). However, these methods are expensive to train due to the evaluation of complex tensor inversions. Recent work by (Barkan, 2016; Bražinskas et al., 2017) show that it is possible to train word embeddings through a variational Bayes (Bishop, 2006) framework.

KG2E (He et al., 2015) proposed a probabilistic embedding method for modelling the uncertainties in KGs. However, this was not a generative model. (Xiao et al., 2016) argued theirs was the first generative model for knowledge graph embeddings. However, their work is empirically worse than a few of the generative models built under our proposed framework, and their method is restricted to a Gaussian distribution prior. In contrast, we can use any prior that permits a re-parameterisation trick — such as a Normal (Kingma & Welling, 2013a) or von-Mises distribution (Davidson et al., 2018).

Later, (Kipf & Welling, 2016b) proposed a generative model for graph embeddings. However, their method lacks scalability as it requires the use of the full adjacency tensor of the graph as input. Moreover, our work differs in that we create a framework for many variational generative models over multi-relational data, rather than just a single generative model over uni-relational data (Kipf & Welling, 2016b; Grover et al., 2018). In a different task of graph generation, similar models have been used on graph inputs, such as variational auto-encoders, to generate full graph structures, such as molecules (Simonovsky & Komodakis, 2018; Liu et al., 2018; De Cao & Kipf, 2018).

Recent work by (Chen et al., 2018) constructed a variational path ranking algorithm, a graph feature model. This work differs from ours for two reasons. Firstly, it does not produce a generative model for knowledge graph embeddings. Secondly, their work is a graph feature model, with the constraint of at most one relation per entity pair, whereas our model is a latent feature model with a theoretical unconstrained limit on the number of existing relationships between a given pair of entities.

## 5 EXPERIMENTS

### EXPERIMENTAL SETUP

We run each experiment over 500 epochs and validate every 50 epochs. Each KB dataset is separated into 80 % training facts, 10% development facts, and 10% test facts. During the evaluation, for each fact, we include every possible corrupted version of the fact under the local closed world assumption, such that the corrupted facts do not exist in the KB. Subsequently, we make a ranking prediction of every fact and its corruptions, summarised by mean rank and filtered hits@m.

During training Bernoulli sampling to estimate the ELBO was used, with linear warm-up (Bowman et al., 2016; Davidson et al., 2018), compression cost (Blundell et al., 2015), ADAM (Kingma & Ba, 2014) Glorot's initialiser for mean vectors (Glorot & Bengio, 2010) and variance values initialised uniformly to embedding size$^{-1}$. We experimented both with a $\mathcal{N}(0,1)$ and a $\mathcal{N}(0, \text{embedding size}^{-1})$ prior on the latent variables.

Table 1 shows definite improvements on WN18 for Variational ComplEx compared with the initially published ComplEX. We believe this is due to the well-balanced model regularisation induced by the zero mean unit variance Gaussian prior. Table 1 also shows that the variational framework is outperformed by existing non-generative models, highlighting that the generative model may be better suited at identifying and predicting symmetric relationships. WordNet18 (Bordes et al., 2013b) (WN18) is a large lexical database of English. WN18RR is a subset with only asymmetric relations. FB15K is a large collaboratively made dataset which covers a vast range of relationships and entities, with FB15K-257 (Toutanova & Chen, 2015), with 257 relations — a significantly reduced number from FB15K due to being a similarly refined asymmetric dataset. We now compare our model to the previous state-of-the-art multi-relational generative model TransG (Xiao et al., 2016), as well as to a previously published probabilistic embedding method KG2E (similarly represents each embedding with a multivariate Gaussian distribution) (He et al., 2015) on the WN18 dataset. Table 2 makes clear the improvements in the performance of the previous state-of-the-art generative multi-relational knowledge graph model. LFM has marginally worse performance than the state-of-the-art model on raw Hits@10. We conjecture two reasons may cause this discrepancy. Firstly, the fact the authors of TransG use negative samples provided only (True negative examples), whereas we generated our negative samples using the local closed world assumption (LCWA) . Secondly, we only use one negative sample per positive to estimate the Evidence Lower Bound using Bernoulli sampling, whereas it is likely they used significantly more negative samples.

| Dataset | Scoring Function | MR | | Hits @ | | |
|---|---|---|---|---|---|---|
| | | Filter | Raw | 1 | 3 | 10 |
| WN18 | V DistMult ( LIM) | 786 | 798 | 0.671 | 0.931 | 0.947 |
| | DistMult | 813 | 827 | 0.754 | 0.911 | 0.939 |
| | V ComplEx ( LIM) | **753** | **765** | 0.934 | **0.945** | **0.952** |
| | ComplEx* | - | - | **0.939** | 0.944 | 0.947 |
| WN18 RR | V DistMult ( LIM) | 6095 | **6109** | 0.357 | 0.423 | 0.440 |
| | DistMult | 8595 | 8595 | 0.367 | 0.390 | 0.412 |
| | V ComplEx ( LFM ) | 6500 | 6514 | 0.385 | 0.446 | 0.489 |
| | ComplEx** | **5261** | - | **0.41** | **0.46** | **0.51** |
| FB15K -257 | V DistMult ( LIM) | 679 | 813 | 0.171 | 0.271 | 0.397 |
| | DistMult | **355** | **501** | **0.187** | **0.282** | 0.400 |
| | V ComplEx ( LIM) | 1221 | 1347 | 0.168 | 0.260 | 0.369 |
| | ComplEx** | 339 | - | 0.159 | 0.258 | **0.417** |

Table 1: Filtered and Mean Rank (MR) for the models tested on the WN18, WN18RR, and FB15K datasets. Hits@m metrics are filtered. Variational written with a "V". *Results reported from (Trouillon et al., 2016b) and **Results reported from (Dettmers et al., 2017) for ComplEx model. "-" in a table cell equates to that statistic being un-reported in the models referenced paper.

| Dataset | Scoring Function | | MR | Raw Hits@ | Filtered Hits @ | | |
|---|---|---|---|---|---|---|---|
| | | Raw | Filter | 10 | 1 | 3 | 10 |
| WN18 | KG2E (He et al., 2015) | **362** | **345** | 0.805 | - | - | 0.932 |
| | TransG (Generative) (Xiao et al., 2016) | 345 | 357 | **0.845** | - | - | 0.949 |
| | Variational ComplEx ( LFM ) | 753 | 765 | 0.836 | **0.934** | **0.945** | **0.952** |

Table 2: Variational Framework vs. Generative Modles

### 5.0.1 LINK PREDICTION ANALYSIS

Section 5.1 and Section 5.2 explores the predictions made by Latent Information Model with ComplEx scoring function, trained with Bernoulli sampling to estimate the ELBO on the WN18RR dataset, then Section 5.3 will analyse the values of embeddings learnt for this task. Lastly, Section 5.3.1 will perform an extrinsic evaluation on learnt embedding representations for the more accessible to interpret Nations dataset.

We split the analysis into the predictions of subject $((?, r, o))$ or object $((s, r, ?))$ for each test fact. Note all results are filtered predictions, i.e., ignoring the predictions made on negative examples generated under LCWA — using a randomly corrupted fact (subject or object corruption) as a negative example.

### 5.1 SUBJECT PREDICTION

Table 3 shows that the relation "_derivationally_related_form", comprising 34% of test subject predictions, was the most accurate relation to predict for Hits@1 when removing the subject from the tested fact. Contrarily, "_member_of_domain_region" with zero Hits@1 subject prediction, making up less than 1% of subject test predictions. However, "_member_meronym " was the least accurate and prominent (8% of the test subject predictions) for subject Hits@1. We learn from this that even for a near state-of-the-art model there is a great deal of improvement to be gained among asymmetric modelling.

### 5.2 OBJECT PREDICTION

Table 4 displays similar results to Table 3, as before the relation "_derivationally_related_form" was the most accurate relation to predict Hits@1. Table 4 differs from Table 3 as it highlights the Latent Information Model's inability to achieve a high Hits@1 performance predicting objects for

|  | Proportion | Hits@1 | Hits@3 | Hits@10 |
|---|---|---|---|---|
| _hypernym | 0.399170 | 0.091926 | 0.123102 | 0.162270 |
| _derivationally_related_form | 0.342693 | 0.947858 | 0.956238 | 0.959032 |
| _member_meronym | 0.080728 | 0.007905 | 0.019763 | 0.035573 |
| _has_part | 0.054882 | 0.011628 | 0.058140 | 0.122093 |
| _instance_hypernym | 0.038928 | 0.393443 | 0.508197 | 0.713115 |
| _synset_domain_topic_of | 0.036375 | 0.219298 | 0.315789 | 0.464912 |
| _also_see | 0.017869 | 0.589286 | 0.625000 | 0.625000 |
| _verb_group | 0.012444 | 0.743590 | 0.974359 | 0.974359 |
| _member_of_domain_region | 0.008296 | 0.000000 | 0.038462 | 0.115385 |
| _member_of_domain_usage | 0.007658 | 0.000000 | 0.000000 | 0.000000 |
| _similar_to | 0.000957 | 1.000000 | 1.000000 | 1.000000 |

Table 3: Latent Information Model with ComplEx: Subject Prediction on WN18RR. Proportion represents the ratio of the positive examples which are of that relation's category.

|  | Proportion | Hits@1 | Hits@3 | Hits@10 |
|---|---|---|---|---|
| _hypernym | 0.399170 | 0.000000 | 0.014388 | 0.046363 |
| _derivationally_related_form | 0.342693 | 0.945996 | 0.957169 | 0.959032 |
| _member_meronym | 0.080728 | 0.031621 | 0.047431 | 0.086957 |
| _has_part | 0.054882 | 0.034884 | 0.081395 | 0.139535 |
| _instance_hypernym | 0.038928 | 0.024590 | 0.081967 | 0.131148 |
| _synset_domain_topic_of | 0.036375 | 0.035088 | 0.043860 | 0.078947 |
| _also_see | 0.017869 | 0.607143 | 0.625000 | 0.625000 |
| _verb_group | 0.012444 | 0.897436 | 0.974359 | 0.974359 |
| _member_of_domain_region | 0.008296 | 0.038462 | 0.076923 | 0.076923 |
| _member_of_domain_usage | 0.007658 | 0.000000 | 0.000000 | 0.000000 |
| _similar_to | 0.000957 | 1.000000 | 1.000000 | 1.000000 |

Table 4: Latent Information Model with ComplEx: Object Prediction on WN18RR

the "_hypernym" relation, which is significantly hindering model performance as it is the most seen relation in the test set— its involvement in 40% of object test predictions.

## 5.3 EMBEDDING ANALYSIS

These results hint at the possibility that the slightly stronger results of WN18 are due to covariances in our variational framework able to capture information about symbol frequencies. We verify this by plotting the mean value of covariance matrices, as a function of the entity or predicate frequencies (Figure 3). The plots confirm our hypothesis: covariances for the variational Latent Information Model grows with the frequency, and hence the LIM would put a preference on predicting relationships between less frequent symbols in the knowledge graph. This also suggests that covariances from the generative framework can capture genuine information about the generality of symbolic representations.

### 5.3.1 EXTRINSIC EVALUATION: VISUAL EMBEDDING ANALYSIS

We project the high dimensional mean embedding vectors to two dimensions using Probabilistic Principal Component Analysis (PPCA) (Tipping & Bishop, 1999) to project the variance embedding vectors down to two dimensions using Non-negative Matrix Factorisation (NNMF) (Févotte & Idier, 2011). Once we have the parameters for a bivariate normal distribution, we then sample from the bivariate normal distribution 1,000 times and then plot a bi-variate kernel density estimate of these samples. By visualising these two-dimensional samples, we can conceive the space in which the entity or relation occupies. We complete this process for the subject, object, relation, and a randomly sampled corrupted entity (under LCWA) to produce a visualisation of a fact, as shown in Figure 4.

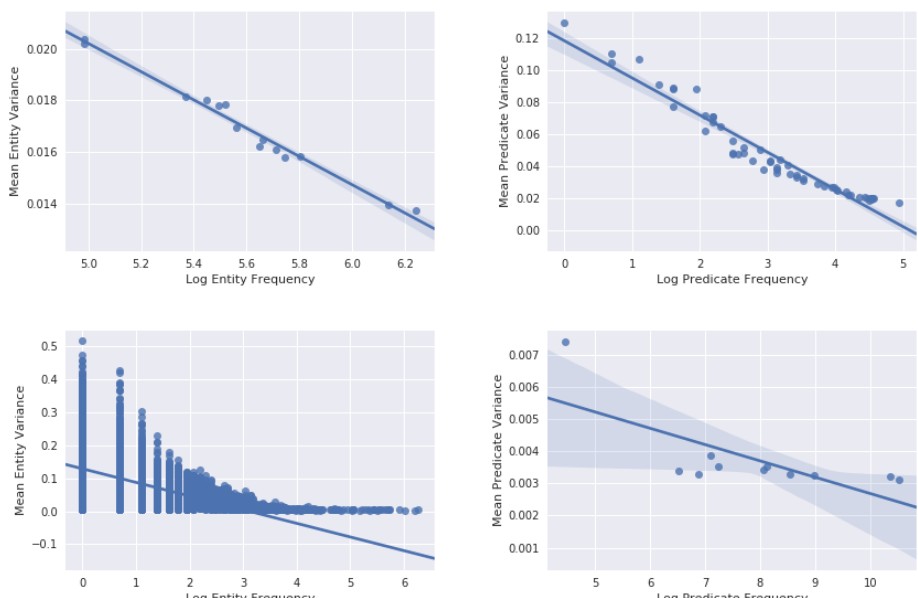

Figure 3: Mean Variance vs. log frequency. From left to right: Nations Entity Analysis, Nations Predicate Analysis, WN18RR Entity Analysis and WN18RR Predicate Analysis.

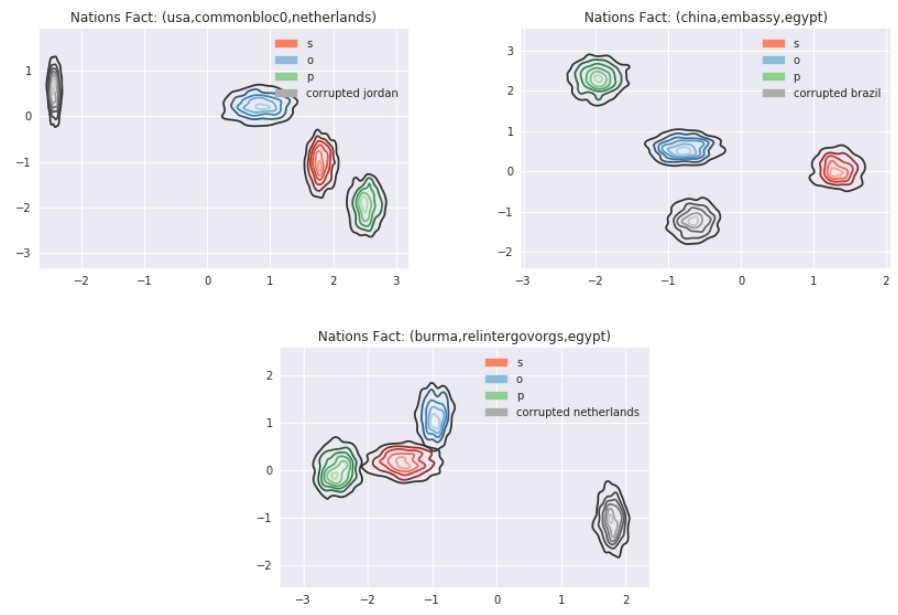

Figure 4: True Positives

Figure 4 displays three true positives from test time predictions. The plots show that the variational framework can learn high dimensional representations which when projected onto lower (more interpretable) dimensions, the distribution over embeddings are shaped to occupy areas at which true facts lie.

Figure 4 displays a clustering of the subject, object, and predicate that create a positive (true) fact. We also observe a separation between the items which generate a fact and a randomly sampled (corrupted) entity which is likely to create a negative (false) fact. The first test fact "(USA, Commonbloc0,

Netherlands)" shows clear irrationality similarity between all objects in the tested fact, i.e. the vectors are pointing towards a south-east direction. We can also see that the corrupted entity Jordan is quite a distance away from the items in the tested fact, which is good as Jordan does not share a common bloc either USA or Netherlands. We used scoring functions which measure the similarity between two vectors, however, for more sophisticated scoring functions which distance/ similarity is not important to the end result we would unlikely see such interpretable images. This analysis of the learnt distributions is evidence to support the notion of learnt probabilistic semantics through using this framework.

### 5.4 Conclusion

We have successfully created a framework allowing a model to learn embeddings of any prior distribution that permits a re-parametrisation trick via any score function that permits maximum likelihood estimation of the scoring parameters. The framework reduces the parameter by one hyper-parameter — as we typically would need to tune a regularisation term for an l1/ l2 loss term, however as the Gaussian distribution is self-regularising this is deemed unnecessary for matching state-of-the-art performance. We have shown, from preliminary experiments, that these display competitive results with current models. Overall, we believe this work will enable knowledge graph researchers to work towards the goal of creating models better able to express their predictive uncertainty.

## 6 Further Work

The score we acquire at test time even through forward sampling does not seem to differ much compared with the mean embeddings, thus using the learnt uncertainty to impact the results positively is a fruitful path. We would also like to see additional exploration into various encoding functions, as we used only the most basic for these experiments. We would also like to see more research into measuring how good the uncertainty estimate is.

### Acknowledgments

We would like to thank all members of the Machine Reading lab for useful discussions.

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

APPENDIX

PROOF: LFM

The marginal distribution over $\mathcal{D}$ is then defined as follows:

$$p^\theta(\mathcal{D}) \triangleq \int \prod_{(\tau, y_\tau) \in \mathcal{D}} p^\theta(\mathbf{h}_\tau) p^\theta(y_\tau \mid \mathbf{h}_\tau) d\mathcal{H} \tag{12}$$

The log-marginal likelihood of the data is the following:

$$\begin{aligned}
\log p^\theta(\mathcal{D}) &= \log \int \prod_{(\tau, y_\tau) \in \mathcal{D}} p^\theta(\mathbf{h}_\tau) p^\theta(y_\tau \mid \mathbf{h}_\tau) d\mathcal{H} \\
&\geq \int \log \prod_{(\tau, y_\tau) \in \mathcal{D}} p^\theta(\mathbf{h}_\tau) p^\theta(y_\tau \mid \mathbf{h}_\tau) d\mathcal{H} \\
&= \int \sum_{(\tau, y_\tau) \in \mathcal{D}} \log p^\theta(\mathbf{h}_\tau) + \log p^\theta(y_\tau \mid \mathbf{h}_\tau) d\mathcal{H} \\
&= \sum_{(\tau, y_\tau) \in \mathcal{D}} \int \log p^\theta(\mathbf{h}_\tau) + \log p^\theta(y_\tau \mid \mathbf{h}_\tau) d\mathbf{h}_\tau \\
&= \sum_{(\tau, y_\tau) \in \mathcal{D}} \text{ELBO}_\tau
\end{aligned} \tag{13}$$

Given a triple $\tau$, the term $\text{ELBO}(\tau)$ can be rewritten as follows:

$$\begin{aligned}
\text{ELBO}_\tau &= \int \log p^\theta(y_\tau \mid \mathbf{h}_\tau) p^\theta(\mathbf{h}_\tau) d\mathbf{h}_\tau \\
&= \int \log \frac{p^\theta(y_\tau \mid \mathbf{h}_\tau) p^\theta(\mathbf{h}_\tau)}{q^\phi(\mathbf{h}_\tau)} q^\phi(\mathbf{h}_\tau) d\mathbf{h}_\tau \\
&= \int \left[ \log p^\theta(y_\tau \mid \mathbf{h}_\tau) + \log p^\theta(\mathbf{h}_\tau) - \log q^\phi(\mathbf{h}_\tau) \right] q^\phi(\mathbf{h}_\tau) d\mathbf{h}_\tau \\
&= \int \log p^\theta(y_\tau \mid \mathbf{h}_\tau) q^\phi(\mathbf{h}_\tau) d\mathbf{h}_\tau + \int q^\phi(\mathbf{h}_\tau) \log \frac{p^\theta(\mathbf{h}_\tau)}{q^\phi(\mathbf{h}_\tau)} d\mathbf{h}_\tau \\
&= \mathbb{E}_{q^\phi} \left[ \log p^\theta(y_\tau \mid \mathbf{h}_\tau) \right] - \text{KL}[q^\phi(\mathbf{h}_\tau) \mid\mid p^\theta(\mathbf{h}_\tau)]
\end{aligned} \tag{14}$$

PROOF: LIM

The marginal distribution over $\mathcal{D}$ is then defined as follows:

$$p^\theta(\mathcal{D}) \triangleq \int \prod_{e \in \mathcal{E}} p^\theta(\mathbf{h}_e) \prod_{\mathrm{p} \in \mathcal{R}} p^\theta(\mathbf{h}_\mathrm{p}) \prod_{(\tau, y_\tau) \in \mathcal{D}} p^\theta(y_\tau \mid \mathbf{h}_\tau) d\mathcal{H} \tag{15}$$

The log-marginal likelihood of the data is the following:

$$\begin{aligned}
\log p^\theta(\mathcal{D}) &= \log \int \prod_{e \in \mathcal{E}} p^\theta(\mathbf{h}_e) \prod_{\mathrm{p} \in \mathcal{R}} p^\theta(\mathbf{h}_\mathrm{p}) \prod_{(\tau, y_\tau) \in \mathcal{D}} p^\theta(y_\tau \mid \mathbf{h}_\tau) d\mathcal{H}_e, \mathcal{H}_\mathrm{p} \\
&= \log \int p^\theta(D|\mathcal{H}_e, \mathcal{H}_\mathrm{p}) p^\theta(\mathcal{H}_e) p^\theta(\mathcal{H}_\mathrm{p}) \, d\mathcal{H}_e, \mathcal{H}_\mathrm{p} \\
&= \log \int p^\theta(D|\mathcal{H}_e, \mathcal{H}_\mathrm{p}) p^\theta(\mathcal{H}_e) \frac{q^\phi(\mathcal{H}_e)}{q^\phi(\mathcal{H}_e)}) p^\theta(\mathcal{H}_\mathrm{p}) \frac{q^\phi(\mathcal{H}_\mathrm{p})}{q^\phi(\mathcal{H}_\mathrm{p})} \, d\mathcal{H}_e, \mathcal{H}_\mathrm{p} \\
&= \geq \mathbb{E}_{q^\phi}[\log(p^\theta(D|\mathcal{H}_e, \mathcal{H}_\mathrm{p}) \frac{p^\theta(\mathcal{H}_e)}{q^\phi(\mathcal{H}_e)} \frac{p^\theta(\mathcal{H}_\mathrm{p})}{q^\phi(\mathcal{H}_\mathrm{p})})] \\
&= \mathbb{E}_{q^\phi}[\log(p^\theta(D|\mathcal{H}_e, \mathcal{H}_\mathrm{p})] - \mathbb{E}_{q^\phi}[\log(\frac{q^\phi(\mathcal{H}_e)}{p^\theta(\mathcal{H}_e)})] - \mathbb{E}_{q^\phi}[\log(\frac{q^\phi(\mathcal{H}_\mathrm{p})}{p^\theta(\mathcal{H}_\mathrm{p})})] \\
&= \mathbb{E}_{q^\phi}\left[\log p^\theta(\mathcal{D} \mid \mathcal{H}_e, \mathcal{H}_\mathrm{p})\right] - \mathrm{KL}[q^\phi(\mathcal{H}_e) \mid\mid p^\theta(\mathcal{H}_e)] - \mathrm{KL}[q^\phi(\mathcal{H}_\mathrm{p}) \mid\mid p^\theta(\mathcal{H}_\mathrm{p})]
\end{aligned} \tag{16}$$

