# OpenReview forum: "Neural Variational Inference For Embedding Knowledge Graphs"
_ICLR.cc/2019/Conference_

### Official Review · AnonReviewer2 · 2018-10-15
**Interesting work, not mature for publication at ICLR**

**Rating:** 4
**Confidence:** 3

**Review:**

The paper presents two variational inference frameworks for generative models of knowledge graphs. Such models are based respectively on latent fact model and latent information model.
The authors argue that with the presented framework the underlying probabilistic semantics can be discovered. Experiments show performances comparable with state-of-art approaches.

Unfortunately, the paper seems to me not clear and rather incomplete in its actual form.
Overall, the proposal is novel. I cannot decide about significance because results do not outperform those of other approaches. To this extent, the authors should better discuss the results, explaining in more detail why this approach should be used instead of others (scales better, is faster, etc.).

In the abstract, it is asserted that one can discover underlying probabilistic semantics, but in the corpus of the paper this aspect is not described or mentioned in detail.
Similar problem for the reference to von-Mises distribution. This distribution is just named, it is said that the framework can handle such a distribution, but a reference to a paper and/or a short paragraph to explain the sentence are missing. This statement now results to be just information disconnected by the rest of the paper.

In a similar way, many other points suffer from a poor organization in my opinion.
When describing LIM an error is introduced here that is then copied and pasted throughout the paper: in the productory on p, p is in R not in E. This is a simple typo, but the fact that it is repeated so many times, also in the proof, gives me the feeling that the paper was written at the last moment.
Figures 1 and 2 are never referred.

Formula 6 must be better explained. If I have not lost something, n is the number of labeled triples, s_c is undefined, b_c is the probability of s_c to be equal to 1, the index i is never used. The paper lacks information here.

As regards the experimental part, some results are shown in subsection 4.3 called link prediction, others in section 5 called link prediction analysis. This organization does not seem to me to be really optimal. I would suggest creating an experimental section.
Moreover, the tests should be better explained, the tables are shown without specifying how they are built and how the values are collected. Information is provided in the appendix but could be included in the paper as the maximum limit is of 10 pages (8 suggested but I think an extra half page can be used).
The knowledge bases used should be at least cited, I know that freebase and wordnet are well-known but somewhere, in the description of the test, the name should be included. Also to specify the characteristics of the versions (WN18 vs WN18RR). Moreover, what does the value -257 in column 1, row 4 means?
Then, it is said that Table 1 shows improvements for ComplEx, but such improvements are rather low, is there a way to prove their significance? Otherwise, I would say that the performance is the same for WN18.
Tables 1 and 2 contain cells with '-' value, what does it mean?

Discussion about table 3 is incomplete in my opinion. First of all, the "proportion" column should be described. Also, on one hand, it is true that the _member_meronym is the least accurate and prominent but the most problem may come from _hypernym, which is the most prominent and the accuracy is also low. This fact is highlighted for table 4 but not for table 3.

Minor issues
- sec 3: references to Miao et al. must be enclosed by brackets
- sec 4.3: "We believe this *is* due to ..."
- sec 5.2: what is Model A? Also, the sentence seems incomplete.

Pros:
- Novel approach

Cons
- Test results are not convincing
- The paper is not mathematically sound
- The paper needs to be re-organized

---

> ### Author Response · Authors · 2018-11-21
> **Consistently outperformed existing generative models and introduced a tool to enable large-scale ELBO estimation tailored for knowledge graphs.**
>
> The authors greatly appreciate the in-depth feedback.
>
> *As previously mentioned the focus of our work is on comparison to prior generative knowledge base models, there we respectfully disagree with the reviewer's statement of “do not outperform those of other approaches”, as we outperformed existing generative models. But the significance of our contribution lies in the first intersection of variational inference techniques applied to knowledge graph link prediction, as well as the novel training techniques implemented to scale these probabilistic systems to large-scale KG’s with ease.
>
> *We believe the underlying probabilistic semantics uncovered are showcased through a visual exploration of the learned probabilistic embedding semantics, which has been added into the updated paper. Thanks to your suggestion, we have also added in a discussion of the use of von-Mises distribution as well as the paper which inspired the idea.
>
> *The index errors have been corrected as well as a reference to Figure 1 and Figure 2 added in their corresponding model derivations. Thank you for pointing them out!
>
> *Formula six now has additional information included, as well as a more straightforward and more specific reformulation to the LFM, Formula seven now expresses the ELBO optimization of LIM with the simplified notation. We have also corrected the proofs in the appendix and now believe the paper to be mathematically sound --- if you disagree please can you specify specific proofs/ equations so we can discuss/ fix the problem?
>
> *We have re-structured the work so that it follows a more typical structure as recommended and included citations to the original datasets and descriptions of what the -257 means (257 relations in the dataset, which is reduced as we only keep the asymmetric relations in order to produce a more challenging dataset). The significance of the results over ComplEX could be proved using confidence margins, however as mentioned earlier we are not focused on beating SotA methods, we are focused on furthering generative knowledge graph research and comparing our method to alternative generative models.
>
> *In Table 1 and 2 the - means that the metric was not reported in the published paper the statistics were referenced from, which I have included in the latex file.
>
> *Proportion in Table 3 represents the ratio of the positive examples which are of that relation's category. The similar points are also now re-iterated across both tables.
>
> *The minor issues are now corrected --- Model A is the Latent Information Model (which was the original name).

---

> > ### Public Comment · (anonymous) · 2018-11-22
> > **Missing discussion on existing variational approaches to graph generative modeling**
> >
> > There are a couple of existing works in generative modeling of graphs with variational approaches that are missing a discussion/comparison in the current work:
> >
> > - GraphVAE https://arxiv.org/pdf/1802.03480.pdf
> > - Graphite https://arxiv.org/abs/1803.10459 (this one explicitly addresses the scalability issue of variational graph autoencoder of Kipf&Welling )

---

> > > ### Author Response · Authors · 2018-11-23
> > > **GraphVAE and Graphite focus only on uni-relational graph structures (already briefly discussed), whereas we focus purely on multi-relational graph structures.**
> > >
> > > We thank you for the anonymous comment. However, the papers suggested are uni-relational model --- which have already been included in our discussion on Kipf's [1] generative graph convolution model (S4). We have now included papers [2,3,4,5] in an additional few sentences on generative graph modeling, however, they are not fit for comparison as they do not function on multi-relational data --- thus do not alter the main contributions of this paper. Secondary differences: GraphVAE is focused on generating the full graph structure, whereas we are focused on generating the sub-structures (such as a link).
> > >
> > > [1] Kipf, T.N. and Welling, M., 2016. Variational graph auto-encoders. arXiv preprint arXiv:1611.07308.
> > > [2] Simonovsky, M. and Komodakis, N., 2018. GraphVAE: Towards Generation of Small Graphs Using Variational Autoencoders. arXiv preprint arXiv:1802.03480.
> > > [3] Grover, A., Zweig, A. and Ermon, S., 2018. Graphite: Iterative generative modeling of graphs. arXiv preprint arXiv:1803.10459.
> > > [4] Liu, Q., Allamanis, M., Brockschmidt, M. and Gaunt, A.L., 2018. Constrained Graph Variational Autoencoders for Molecule Design. arXiv preprint arXiv:1805.09076.
> > > [5] De Cao, N. and Kipf, T., 2018. MolGAN: An implicit generative model for small molecular graphs. arXiv preprint arXiv:1805.11973.

---

> > ### Comment · AnonReviewer2 · 2018-11-23
> > **Quality has been improved**
> >
> > I find the current version's quality has been improved and I am happy with the answers from the authors, so I am willing to increase my score toward acceptance. However, it seems that more work is needed to have a paper mature enough for the conference (I am thinking on the missing references cited by an anonymous commenter), therefore I would wait the other reviewers comments before deciding how much to increase my score.

---

> > > ### Author Response · Authors · 2018-11-23
> > > **Missing citation issue already addressed**
> > >
> > > Dear Reviewer,
> > >
> > > We thank you for the promising feedback. We have responded to the anonymous comment recently around missing comparisons and stressed the missing references are not crucial to the papers story —- as we focus on different tasks (uni vs multi relational data), they are instead only additional related work, which was already introduced in the original submission when I discussed Kipf’s generative model “Variational graph networks”.
> > >
> > > I hope this no longer remains an issue, and appreciate the fact you wish to wait longer before altering the score.
> > >
> > > Thanks again for your time.

---

### Official Review · AnonReviewer1 · 2018-10-31
**Interesting variational framework, results not convincing enough**

**Rating:** 5
**Confidence:** 5

**Review:**

* The paper proposed a neural variational inference framework for knowledge graph embedding. The paper proposed two models (Latent Fact Model and Latent Information Model) where the neural variational inference is carried out, with competitive results on standard datasets (WN18 and FB15K).

* I am not fully convinced of the advantage of this variational inference approach compared with the optimization approach used in TransE, TransG, DistMult and ComplEx. As can be seen in Table 1, the best performance on WN18-RR and FB15K-257 are obtained without variational inference. In Table 2, performance of the variational inference approach is not as good as other approaches under the MR or Raw Hits metrics. Moreover, performance on FB15K is not reported in Table 2, which makes the result not as complete as Table 1.

* It seems that the main difference between Latent Fact Model and Latent Information Model is the way prior is imposed on h. The authors may want to explain in plain language the differences and the motivation behind that.

* It is unclear how the (\tau,y) labeled triples are generated, especially for the negative examples with y=0. Is it obtained by randomly corrupting the triples in the knowledge base, as done in other work? It would be better to make this point clear.

---

> ### Author Response · Authors · 2018-11-21
> **The focus of our work is on comparison to prior generative knowledge base models, and on the theoretical tools required.**
>
> Thank you for the detailed feedback. Regarding the raised issues;
>
> *The focus of our work is on comparison to prior generative knowledge base models, not on beating SotA neural link prediction methods. We have shown increased performance against previous generative approaches while laying future support for incorporating variational inference techniques with statistical relational learning. That lack of FB15K results in Table 2 was due to the FB15K results not being reported in the referenced articles.
>
> *We have now added in a more explicit description of the underlying assumptions of each model and a brief discussion of the model differences. In essence, the independent assumption on latent variables in LIM leads to a significantly simpler to compute ELBO, requiring fewer approximations to be made. In contrast, the LFM retains dependence across the latent variables.
>
> *We have included more information as to how negative samples are generated --- in prior work [1,2,3,4] this is commonly done by corrupting a positive example. The additional information is in the experimental setup as well as in the section on link prediction (Section 5.1).
>
>
> References
>
> [1] Dettmers, T., Minervini, P., Stenetorp, P. and Riedel, S., 2017. Convolutional 2d knowledge graph embeddings. arXiv preprint arXiv:1707.01476.
>
> [2] Nickel, M., Rosasco, L. and Poggio, T.A., 2016, February. Holographic Embeddings of Knowledge Graphs. In AAAI (Vol. 2, No. 1, pp. 3-2).
>
> [3] Trouillon, T., Dance, C.R., Gaussier, É., Welbl, J., Riedel, S. and Bouchard, G., 2017. Knowledge graph completion via complex tensor factorization. The Journal of Machine Learning Research, 18(1), pp.4735-4772.
>
> [4] Yang, B., Yih, W.T., He, X., Gao, J. and Deng, L., 2014. Embedding entities and relations for learning and inference in knowledge bases. arXiv preprint arXiv:1412.6575.

---

### Official Review · AnonReviewer3 · 2018-11-04
**This work proposed variational embedding for knowledge graph inference tasks. However, neither the methods nor the results are really impressive**

**Rating:** 5
**Confidence:** 3

**Review:**


This work proposed two variational embedding methods for knowledge graph inference tasks. The experiments show slight improvements compared to other variational embedding methods, e.g., KG2E, TransG, and slight improvement on the WN18 dataset compared with the non-variational method. On the other hand, both the embedding method and the training of variational models used in this work are already well developed. Thus, this work doesn’t show too much novel contribution. However, the reviewer really appreciate the visualization provided in Fig. 4.
Minor issues:
1)	Notation of the KL divergence is not conventional
2)	There are some mistakes of indices for predicates, e.g., in Eq. 7, 8.

---

> ### Author Response · Authors · 2018-11-21
> **We respectfully disagree with the claim that our work does not show much novel contribution.**
>
> We thank the reviewer for their comments. We respectfully disagree with the claim that our work does not show too much novel contribution. The probabilistic graphical model approach to knowledge graphs, as well as the training method using Bernoulli sampling to estimate the evidence lower bound, is novel. This novel ELBO approximation will be a crucial technique to scale VI for knowledge graph research as we found using only negative sampling (instead of using the ELBO or ELBO approximation), a typical trick to reduce the negative training examples, leads to a highly unstable training process. The technique trades off exact ELBO computation with an approximation, which is of orders of magnitude faster. Secondly, this ELBO approximation for LIM differs from typical stochastic variational inference techniques, as we are able to compute the full KL divergence term and are only required to approximate the full batches expected likelihood. Our work provides a framework which allows the specification of a family of generative knowledge graph models with any scoring functions that permits MLE of their parameters and any distribution that permits a reparameterization trick, allowing further research in VI to be easily incorporated into knowledge graph modeling.
>
> We thank the reviewer for the corrections—we corrected the KL divergence notation as well as the predicate index errors.

---

### Meta-Review · Area_Chair1 · 2018-12-08
**Intersting work with slighlty limited originality that would benefit from a clearer motivation.**

**Confidence:** 4
**Recommendation:** Reject

**Metareview:**

The paper proposes a novel variational inference framework for knowledge graphs which is evaluated on link prediction benchmark sets and is competitive to previous generative approaches.
While the idea is interstnig and technically correct, the originality of the contribution is limited,
and the paper would be clearly improved by providing a clearer motivation for using generative models instead of standard methods and a experimental demonstration of  the benefits of using a generative instead of a discriminative model,  especially since the standard method perform slightly better in the experiments. Overall, the work is slightly under the acceptance threshold.